# Learning Combinatorial Optimization Algorithms over Graphs

**Hanjun Dai**[†*]**, Elias B. Khalil**[†*]**, Yuyu Zhang**[†]**, Bistra Dilkina**[†]**, Le Song**[†§]
[†] College of Computing, Georgia Institute of Technology
[§] Ant Financial
{hanjun.dai, elias.khalil, yuyu.zhang, bdilkina, lsong}@cc.gatech.edu

## Abstract

The design of good heuristics or approximation algorithms for NP-hard combinatorial optimization problems often requires significant specialized knowledge and trial-and-error. Can we automate this challenging, tedious process, and learn the algorithms instead? In many real-world applications, it is typically the case that the same optimization problem is solved again and again on a regular basis, maintaining the same problem structure but differing in the data. This provides an opportunity for learning heuristic algorithms that exploit the structure of such recurring problems. In this paper, we propose a unique combination of reinforcement learning and graph embedding to address this challenge. The learned greedy policy behaves like a meta-algorithm that incrementally constructs a solution, and the action is determined by the output of a graph embedding network capturing the current state of the solution. We show that our framework can be applied to a diverse range of optimization problems over graphs, and learns effective algorithms for the Minimum Vertex Cover, Maximum Cut and Traveling Salesman problems.

## 1   Introduction

Combinatorial optimization problems over graphs arising from numerous application domains, such as social networks, transportation, telecommunications and scheduling, are NP-hard, and have thus attracted considerable interest from the theory and algorithm design communities over the years. In fact, of Karp's 21 problems in the seminal paper on reducibility [19], 10 are decision versions of graph optimization problems, while most of the other 11 problems, such as set covering, can be naturally formulated on graphs. Traditional approaches to tackling an NP-hard graph optimization problem have three main flavors: exact algorithms, approximation algorithms and heuristics. Exact algorithms are based on enumeration or branch-and-bound with an integer programming formulation, but may be prohibitive for large instances. On the other hand, polynomial-time approximation algorithms are desirable, but may suffer from weak optimality guarantees or empirical performance, or may not even exist for inapproximable problems. Heuristics are often fast, effective algorithms that lack theoretical guarantees, and may also require substantial problem-specific research and trial-and-error on the part of algorithm designers.

All three paradigms seldom exploit a common trait of real-world optimization problems: instances of the same type of problem are solved again and again on a regular basis, maintaining the same combinatorial structure, but differing mainly in their data. That is, in many applications, values of the coefficients in the objective function or constraints can be thought of as being sampled from the same underlying distribution. For instance, an advertiser on a social network targets a limited set of users with ads, in the hope that they spread them to their neighbors; such covering instances need to be solved repeatedly, since the influence pattern between neighbors may be different each time. Alternatively, a package delivery company routes trucks on a daily basis in a given city; thousands of similar optimizations need to be solved, since the underlying demand locations can differ.

---

[*]Both authors contributed equally to the paper.

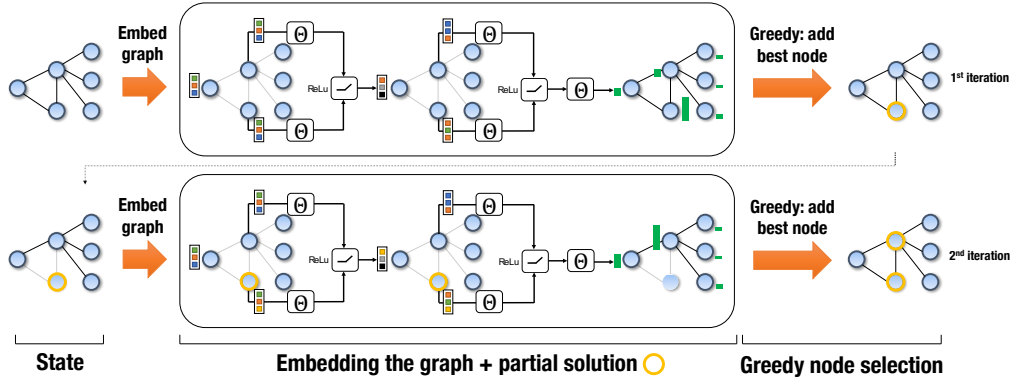

Figure 1: Illustration of the proposed framework as applied to an instance of Minimum Vertex Cover. The middle part illustrates two iterations of the graph embedding, which results in node scores (green bars).

Despite the inherent similarity between problem instances arising in the same domain, classical algorithms do not systematically exploit this fact. However, in industrial settings, a company may be willing to invest in upfront, offline computation and learning if such a process can speed up its real-time decision-making and improve its quality. This motivates the main problem we address:

> **Problem Statement:** Given a graph optimization problem $G$ and a distribution $\mathbb{D}$ of problem instances, can we learn better heuristics that generalize to unseen instances from $\mathbb{D}$?

Recently, there has been some seminal work on using deep architectures to learn heuristics for combinatorial problems, including the Traveling Salesman Problem [37, 6, 14]. However, the architectures used in these works are generic, not yet effectively reflecting the combinatorial structure of graph problems. As we show later, these architectures often require a huge number of instances in order to learn to generalize to new ones. Furthermore, existing works typically use the policy gradient for training [6], a method that is not particularly sample-efficient. While the methods in [37, 6] can be used on graphs with different sizes – a desirable trait – they require manual, ad-hoc input/output engineering to do so (e.g. padding with zeros).

In this paper, we address the challenge of learning algorithms for graph problems using a unique combination of reinforcement learning and graph embedding. The learned policy behaves like a meta-algorithm that incrementally constructs a solution, with the action being determined by a graph embedding network over the current state of the solution. More specifically, our proposed solution framework is different from previous work in the following aspects:

**1. Algorithm design pattern.** We will adopt a *greedy* meta-algorithm design, whereby a feasible solution is constructed by successive addition of nodes based on the graph structure, and is maintained so as to satisfy the problem's graph constraints. Greedy algorithms are a popular pattern for designing approximation and heuristic algorithms for graph problems. As such, the same high-level design can be seamlessly used for different graph optimization problems.

**2. Algorithm representation.** We will use a *graph embedding* network, called `structure2vec` (S2V) [9], to represent the policy in the greedy algorithm. This novel deep learning architecture over the instance graph "featurizes" the nodes in the graph, capturing the properties of a node in the context of its graph neighborhood. This allows the policy to discriminate among nodes based on their usefulness, and generalizes to problem instances of different sizes. This contrasts with recent approaches [37, 6] that adopt a graph-agnostic sequence-to-sequence mapping that does not fully exploit graph structure.

**3. Algorithm training.** We will use fitted $Q$-learning to learn a greedy policy that is parametrized by the graph embedding network. The framework is set up in such a way that the policy will aim to optimize the objective function of the original problem instance *directly*. The main advantage of this approach is that it can deal with delayed rewards, which here represent the remaining increase in objective function value obtained by the greedy algorithm, in a data-efficient way; in each step of the greedy algorithm, the graph embeddings are updated according to the partial solution to reflect new knowledge of the benefit of *each node* to the final objective value. In contrast, the policy gradient approach of [6] updates the model parameters only once w.r.t. the whole solution (e.g. the tour in TSP).

The application of a greedy heuristic learned with our framework is illustrated in Figure 1. To demonstrate the effectiveness of the proposed framework, we apply it to three extensively studied graph optimization problems. Experimental results show that our framework, a single meta-learning algorithm, efficiently learns effective heuristics for each of the problems. Furthermore, we show that our learned heuristics preserve their effectiveness even when used on graphs much larger than the ones they were trained on. Since many combinatorial optimization problems, such as the set covering problem, can be explicitly or implicitly formulated on graphs, we believe that our work opens up a new avenue for graph algorithm design and discovery with deep learning.

## 2 Common Formulation for Greedy Algorithms on Graphs

We will illustrate our framework using three optimization problems over weighted graphs. Let $G(V, E, w)$ denote a weighted graph, where $V$ is the set of nodes, $E$ the set of edges and $w : E \to \mathbb{R}^+$ the edge weight function, i.e. $w(u, v)$ is the weight of edge $(u, v) \in E$. These problems are:

- **Minimum Vertex Cover (MVC):** Given a graph $G$, find a subset of nodes $S \subseteq V$ such that every edge is covered, i.e. $(u, v) \in E \Leftrightarrow u \in S$ or $v \in S$, and $|S|$ is minimized.
- **Maximum Cut (MAXCUT):** Given a graph $G$, find a subset of nodes $S \subseteq V$ such that the weight of the cut-set $\sum_{(u,v) \in C} w(u, v)$ is maximized, where cut-set $C \subseteq E$ is the set of edges with one end in $S$ and the other end in $V \setminus S$.
- **Traveling Salesman Problem (TSP):** Given a set of points in 2-dimensional space, find a tour of minimum total weight, where the corresponding graph $G$ has the points as nodes and is fully connected with edge weights corresponding to distances between points; a tour is a cycle that visits each node of the graph *exactly* once.

We will focus on a popular pattern for designing approximation and heuristic algorithms, namely a greedy algorithm. A greedy algorithm will construct a solution by sequentially adding nodes to a partial solution $S$, based on maximizing some *evaluation function* $Q$ that measures the quality of a node in the context of the current partial solution. We will show that, despite the diversity of the combinatorial problems above, greedy algorithms for them can be expressed using a common formulation. Specifically:

1. A problem instance $G$ of a given optimization problem is sampled from a distribution $\mathbb{D}$, i.e. the $V$, $E$ and $w$ of the instance graph $G$ are generated according to a model or real-world data.
2. A partial solution is represented as an ordered list $S = (v_1, v_2, \ldots, v_{|S|})$, $v_i \in V$, and $\overline{S} = V \setminus S$ the set of candidate nodes for addition, conditional on $S$. Furthermore, we use a vector of binary decision variables $x$, with each dimension $x_v$ corresponding to a node $v \in V$, $x_v = 1$ if $v \in S$ and 0 otherwise. One can also view $x_v$ as a tag or extra feature on $v$.
3. A maintenance (or helper) procedure $h(S)$ will be needed, which maps an ordered list $S$ to a combinatorial structure satisfying the specific constraints of a problem.
4. The quality of a partial solution $S$ is given by an objective function $c(h(S), G)$ based on the combinatorial structure $h$ of $S$.
5. A generic greedy algorithm selects a node $v$ to add next such that $v$ maximizes an evaluation function, $Q(h(S), v) \in \mathbb{R}$, which depends on the combinatorial structure $h(S)$ of the current partial solution. Then, the partial solution $S$ will be extended as

$$S := (S, v^*), \quad \text{where} \quad v^* := \text{argmax}_{v \in \overline{S}} \ Q(h(S), v), \tag{1}$$

and $(S, v^*)$ denotes appending $v^*$ to the end of a list $S$. This step is repeated until a termination criterion $t(h(S))$ is satisfied.

In our formulation, we assume that the distribution $\mathbb{D}$, the helper function $h$, the termination criterion $t$ and the cost function $c$ are all given. Given the above abstract model, various optimization problems can be expressed by using different helper functions, cost functions and termination criteria:

- **MVC:** The helper function does not need to do any work, and $c(h(S), G) = -|S|$. The termination criterion checks whether all edges have been covered.
- **MAXCUT:** The helper function divides $V$ into two sets, $S$ and its complement $\overline{S} = V \setminus S$, and maintains a cut-set $C = \{(u, v) \mid (u, v) \in E, u \in S, v \in \overline{S}\}$. Then, the cost is $c(h(S), G) = \sum_{(u,v) \in C} w(u, v)$, and the termination criterion does nothing.
- **TSP:** The helper function will maintain a tour according to the order of the nodes in $S$. The simplest way is to append nodes to the end of partial tour in the same order as $S$. Then the cost $c(h(S), G) = -\sum_{i=1}^{|S|-1} w(S(i), S(i+1)) - w(S(|S|), S(1))$, and the termination criterion is

activated when $S = V$. Empirically, inserting a node $u$ in the partial tour at the position which increases the tour length the least is a better choice. We adopt this insertion procedure as a helper function for TSP.

An estimate of the quality of the solution resulting from adding a node to partial solution $S$ will be determined by the *evaluation function* $Q$, which will be learned using a collection of problem instances. This is in contrast with traditional greedy algorithm design, where the *evaluation function* $Q$ is typically hand-crafted, and requires substantial problem-specific research and trial-and-error. In the following, we will design a powerful deep learning parameterization for the evaluation function, $\widehat{Q}(h(S), v; \Theta)$, with parameters $\Theta$.

## 3 Representation: Graph Embedding

Since we are optimizing over a graph $G$, we expect that the evaluation function $\widehat{Q}$ should take into account the current partial solution $S$ as it maps to the graph. That is, $x_v = 1$ for all nodes $v \in S$, and the nodes are connected according to the graph structure. Intuitively, $\widehat{Q}$ should summarize the state of such a "tagged" graph $G$, and figure out the value of a new node if it is to be added in the context of such a graph. Here, both the state of the graph and the context of a node $v$ can be very complex, hard to describe in closed form, and may depend on complicated statistics such as global/local degree distribution, triangle counts, distance to tagged nodes, etc. In order to represent such complex phenomena over combinatorial structures, we will leverage a deep learning architecture over graphs, in particular the `structure2vec` of [9], to parameterize $\widehat{Q}(h(S), v; \Theta)$.

### 3.1 Structure2Vec

We first provide an introduction to `structure2vec`. This graph embedding network will compute a $p$-dimensional feature embedding $\mu_v$ for each node $v \in V$, given the current partial solution $S$. More specifically, `structure2vec` defines the network architecture recursively according to an input graph structure $G$, and the computation graph of `structure2vec` is inspired by graphical model inference algorithms, where node-specific tags or features $x_v$ are aggregated recursively according to $G$'s graph topology. After a few steps of recursion, the network will produce a new embedding for each node, taking into account both graph characteristics and long-range interactions between these node features. One variant of the `structure2vec` architecture will initialize the embedding $\mu_v^{(0)}$ at each node as 0, and for all $v \in V$ update the embeddings synchronously at each iteration as

$$\mu_v^{(t+1)} \leftarrow F\left(x_v, \{\mu_u^{(t)}\}_{u \in \mathcal{N}(v)}, \{w(v,u)\}_{u \in \mathcal{N}(v)}; \Theta\right), \tag{2}$$

where $\mathcal{N}(v)$ is the set of neighbors of node $v$ in graph $G$, and $F$ is a generic nonlinear mapping such as a neural network or kernel function.

Based on the update formula, one can see that the embedding update process is carried out based on the graph topology. A new round of embedding sweeping across the nodes will start only after the embedding update for all nodes from the previous round has finished. It is easy to see that the update also defines a process where the node features $x_v$ are propagated to other nodes via the nonlinear propagation function $F$. Furthermore, the more update iterations one carries out, the farther away the node features will propagate and get aggregated nonlinearly at distant nodes. In the end, if one terminates after $T$ iterations, each node embedding $\mu_v^{(T)}$ will contain information about its $T$-hop neighborhood as determined by graph topology, the involved node features and the propagation function $F$. An illustration of two iterations of graph embedding can be found in Figure 1.

### 3.2 Parameterizing $\widehat{Q}(h(S), v; \Theta)$

We now discuss the parameterization of $\widehat{Q}(h(S), v; \Theta)$ using the embeddings from `structure2vec`. In particular, we design $F$ to update a $p$-dimensional embedding $\mu_v$ as:

$$\mu_v^{(t+1)} \leftarrow \text{relu}\left(\theta_1 x_v + \theta_2 \sum_{u \in \mathcal{N}(v)} \mu_u^{(t)} + \theta_3 \sum_{u \in \mathcal{N}(v)} \text{relu}(\theta_4 \, w(v,u))\right), \tag{3}$$

where $\theta_1 \in \mathbb{R}^p$, $\theta_2, \theta_3 \in \mathbb{R}^{p \times p}$ and $\theta_4 \in \mathbb{R}^p$ are the model parameters, and relu is the rectified linear unit $(\text{relu}(z) = \max(0, z))$ applied elementwise to its input. The summation over neighbors is one way of aggregating neighborhood information that is invariant to permutations over neighbors. For simplicity of exposition, $x_v$ here is a binary scalar as described earlier; it is straightforward to extend $x_v$ to a vector representation by incorporating any additional useful node information. To make the

nonlinear transformations more powerful, we can add some more layers of relu before we pool over the neighboring embeddings $\mu_u$.

Once the embedding for each node is computed after $T$ iterations, we will use these embeddings to define the $\widehat{Q}(h(S), v; \Theta)$ function. More specifically, we will use the embedding $\mu_v^{(T)}$ for node $v$ and the pooled embedding over the entire graph, $\sum_{u \in V} \mu_u^{(T)}$, as the surrogates for $v$ and $h(S)$, respectively, i.e.

$$\widehat{Q}(h(S), v; \Theta) = \theta_5^\top \operatorname{relu}([\theta_6 \sum_{u \in V} \mu_u^{(T)}, \theta_7 \mu_v^{(T)}]) \tag{4}$$

where $\theta_5 \in \mathbb{R}^{2p}$, $\theta_6, \theta_7 \in \mathbb{R}^{p \times p}$ and $[\cdot, \cdot]$ is the concatenation operator. Since the embedding $\mu_u^{(T)}$ is computed based on the parameters from the graph embedding network, $\widehat{Q}(h(S), v)$ will depend on a collection of 7 parameters $\Theta = \{\theta_i\}_{i=1}^7$. The number of iterations $T$ for the graph embedding computation is usually small, such as $T = 4$.

The parameters $\Theta$ will be learned. Previously, [9] required a ground truth label for every input graph $G$ in order to train the structure2vec architecture. There, the output of the embedding is linked with a softmax-layer, so that the parameters can by trained end-to-end by minimizing the cross-entropy loss. This approach is not applicable to our case due to the lack of training labels. Instead, we train these parameters together *end-to-end* using reinforcement learning.

## 4 Training: Q-learning

We show how reinforcement learning is a natural framework for learning the evaluation function $\widehat{Q}$. The definition of the evaluation function $\widehat{Q}$ naturally lends itself to a *reinforcement learning* (RL) formulation [36], and we will use $\widehat{Q}$ as a model for the state-value function in RL. We note that we would like to learn a function $\widehat{Q}$ *across a set of $m$ graphs from distribution $\mathbb{D}$, $\mathcal{D} = \{G_i\}_{i=1}^m$*, with potentially different sizes. The advantage of the graph embedding parameterization in our previous section is that we can deal with different graph instances and sizes seamlessly.

### 4.1 Reinforcement learning formulation

We define the states, actions and rewards in the reinforcement learning framework as follows:

1. *States*: a state $S$ is a sequence of actions (nodes) on a graph $G$. Since we have already represented nodes in the tagged graph with their embeddings, the state is a vector in $p$-dimensional space, $\sum_{v \in V} \mu_v$. It is easy to see that this embedding representation of the state can be used across different graphs. The terminal state $\widehat{S}$ will depend on the problem at hand;
2. *Transition*: transition is deterministic here, and corresponds to tagging the node $v \in G$ that was selected as the last action with feature $x_v = 1$;
3. *Actions*: an action $v$ is a node of $G$ that is not part of the current state $S$. Similarly, we will represent actions as their corresponding $p$-dimensional node embedding $\mu_v$, and such a definition is applicable across graphs of various sizes;
4. *Rewards*: the reward function $r(S, v)$ at state $S$ is defined as the change in the cost function after taking action $v$ and transitioning to a new state $S' := (S, v)$. That is,

$$r(S, v) = c(h(S'), G) - c(h(S), G), \tag{5}$$

and $c(h(\emptyset), G) = 0$. As such, the *cumulative reward* $R$ of a terminal state $\widehat{S}$ coincides exactly with the objective function value of the $\widehat{S}$, i.e. $R(\widehat{S}) = \sum_{i=1}^{|\widehat{S}|} r(S_i, v_i)$ is equal to $c(h(\widehat{S}), G)$;
5. *Policy*: based on $\widehat{Q}$, a deterministic greedy policy $\pi(v|S) := \operatorname{argmax}_{v' \in \overline{S}} \widehat{Q}(h(S), v')$ will be used. Selecting action $v$ corresponds to adding a node of $G$ to the current partial solution, which results in collecting a reward $r(S, v)$.

Table 1 shows the instantiations of the reinforcement learning framework for the three optimization problems considered herein. We let $Q^*$ denote the *optimal* Q-function for each RL problem. Our graph embedding parameterization $\widehat{Q}(h(S), v; \Theta)$ from Section 3 will then be a function approximation model for it, which will be learned via $n$-step Q-learning.

### 4.2 Learning algorithm

In order to perform end-to-end learning of the parameters in $\widehat{Q}(h(S), v; \Theta)$, we use a combination of $n$-step Q-learning [36] and *fitted Q-iteration* [33], as illustrated in Algorithm 1. We use the term

Table 1: Definition of reinforcement learning components for each of the three problems considered.

| Problem | State | Action | Helper function | Reward | Termination |
|---|---|---|---|---|---|
| MVC | subset of nodes selected so far | add node to subset | None | -1 | all edges are covered |
| MAXCUT | subset of nodes selected so far | add node to subset | None | change in cut weight | cut weight cannot be improved |
| TSP | partial tour | grow tour by one node | Insertion operation | change in tour cost | tour includes all nodes |

*episode* to refer to a complete sequence of node additions starting from an empty solution, and until termination; a *step* within an episode is a single action (node addition).

Standard (1-step) Q-learning updates the function approximator's parameters *at each step* of an episode by performing a gradient step to minimize the squared loss:

$$(y - \widehat{Q}(h(S_t), v_t; \Theta))^2, \tag{6}$$

where $y = \gamma \max_{v'} \widehat{Q}(h(S_{t+1}), v'; \Theta) + r(S_t, v_t)$ for a non-terminal state $S_t$. The $n$-step Q-learning helps deal with the issue of *delayed rewards*, where the final reward of interest to the agent is only received far in the future during an episode. In our setting, the final objective value of a solution is only revealed after many node additions. As such, the 1-step update may be too myopic. A natural extension of 1-step Q-learning is to wait $n$ steps before updating the approximator's parameters, so as to collect a more accurate estimate of the future rewards. Formally, the update is over the same squared loss (6), but with a different target, $y = \sum_{i=0}^{n-1} r(S_{t+i}, v_{t+i}) + \gamma \max_{v'} \widehat{Q}(h(S_{t+n}), v'; \Theta)$. The fitted Q-iteration approach has been shown to result in faster learning convergence when using a neural network as a function approximator [33, 28], a property that also applies in our setting, as we use the embedding defined in Section 3.2. Instead of updating the Q-function sample-by-sample as in Equation (6), the fitted Q-iteration approach uses *experience replay* to update the function approximator with a batch of samples from a dataset $E$, rather than the single sample being currently experienced. The dataset $E$ is populated during previous episodes, such that at step $t + n$, the tuple $(S_t, a_t, R_{t,t+n}, S_{t+n})$ is added to $E$, with $R_{t,t+n} = \sum_{i=0}^{n-1} r(S_{t+i}, a_{t+i})$. Instead of performing a gradient step in the loss of the current sample as in (6), stochastic gradient descent updates are performed on a random sample of tuples drawn from $E$.

It is known that *off-policy* reinforcement learning algorithms such as Q-learning can be more sample efficient than their policy gradient counterparts [15]. This is largely due to the fact that policy gradient methods require *on-policy* samples for the new policy obtained after each parameter update of the function approximator.

---

**Algorithm 1 Q-learning for the Greedy Algorithm**

---

1: Initialize experience replay memory $\mathcal{M}$ to capacity $N$
2: **for** episode $e = 1$ **to** $L$ **do**
3:     Draw graph $G$ from distribution $\mathbb{D}$
4:     Initialize the state to empty $S_1 = ()$
5:     **for** step $t = 1$ **to** $T$ **do**
6:         $v_t = \begin{cases} \text{random node } v \in \overline{S}_t, & \text{w.p. } \epsilon \\ \text{argmax}_{v \in \overline{S}_t} \widehat{Q}(h(S_t), v; \Theta), & \text{otherwise} \end{cases}$
7:         Add $v_t$ to partial solution: $S_{t+1} := (S_t, v_t)$
8:         **if** $t \geq n$ **then**
9:             Add tuple $(S_{t-n}, v_{t-n}, R_{t-n,t}, S_t)$ to $\mathcal{M}$
10:             Sample random batch from $B \overset{iid.}{\sim} \mathcal{M}$
11:             Update $\Theta$ by SGD over (6) for $B$
12:         **end if**
13:     **end for**
14: **end for**
15: return $\Theta$

---

## 5 Experimental Evaluation

**Instance generation.** To evaluate the proposed method against other approximation/heuristic algorithms and deep learning approaches, we generate graph instances for each of the three problems. For the MVC and MAXCUT problems, we generate Erdős-Renyi (ER) [11] and Barabasi-Albert (BA) [1] graphs which have been used to model many real-world networks. For a given range on the number of nodes, e.g. 50-100, we first sample the number of nodes uniformly at random from that

range, then generate a graph according to either ER or BA. For the two-dimensional TSP problem, we use an instance generator from the DIMACS TSP Challenge [18] to generate uniformly random or clustered points in the 2-D grid. We refer the reader to the Appendix D.1 for complete details on instance generation. We have also tackled the Set Covering Problem, for which the description and results are deferred to Appendix B.

**Structure2Vec Deep Q-learning.** For our method, S2V-DQN, we use the graph representations and hyperparameters described in Appendix D.4. The hyperparameters are selected via preliminary results on small graphs, and then fixed for large ones. Note that for TSP, where the graph is fully-connected, we build the $K$-nearest neighbor graph ($K = 10$) to scale up to large graphs. For MVC, where we train the model on graphs with up to 500 nodes, we use the model trained on small graphs as initialization for training on larger ones. We refer to this trick as "pre-training", which is illustrated in Figure D.2.

**Pointer Networks with Actor-Critic.** We compare our method to a method, based on Recurrent Neural Networks (RNNs), which does not make full use of graph structure [6]. We implement and train their algorithm (PN-AC) for all three problems. The original model only works on the Euclidian TSP problem, where each node is represented by its $(x, y)$ coordinates, and is not designed for problems with graph structure. To handle other graph problems, we describe each node by its adjacency vector instead of coordinates. To handle different graph sizes, we use a singular value decomposition (SVD) to obtain a rank-8 approximation for the adjacency matrix, and use the low-rank embeddings as inputs to the pointer network.

**Baseline Algorithms.** Besides the PN-AC, we also include powerful approximation or heuristic algorithms from the literature. These algorithms are specifically designed for each type of problem:

- **MVC:** *MVCApprox* iteratively selects an uncovered edge and adds both of its endpoints [30]. We designed a stronger variant, called *MVCApprox-Greedy*, that greedily picks the uncovered edge with maximum sum of degrees of its endpoints. Both algorithms are 2-approximations.
- **MAXCUT:** We include *MaxcutApprox*, which maintains the cut set $(S, V \setminus S)$ and moves a node from one side to the other side of the cut if that operation results in cut weight improvement [25]. To make *MaxcutApprox* stronger, we greedily move the node that results in the largest improvement in cut weight. A randomized, non-greedy algorithm, referred to as SDP, is also implemented based on [12]; 100 solutions are generated for each graph, and the best one is taken.
- **TSP:** We include the following approximation algorithms: Minimum Spanning Tree (MST), Farthest insertion (Farthest), Cheapest insertion (Cheapest), Closest insertion (Closest), Christofides and 2-opt. We also add the Nearest Neighbor heuristic (Nearest); see [4] for algorithmic details.

**Details on Validation and Testing.** For S2V-DQN and PN-AC, we use a CUDA K80-enabled cluster for training and testing. Training convergence for S2V-DQN is discussed in Appendix D.6. S2V-DQN and PN-AC use 100 held-out graphs for validation, and we report the test results on another 1000 graphs. We use CPLEX[17] to get optimal solutions for MVC and MAXCUT, and Concorde [3] for TSP (details in Appendix D.1). All approximation ratios reported in the paper are with respect to the best (possibly optimal) solution found by the solvers within 1 hour. For MVC, we vary the training and test graph sizes in the ranges {15–20, 40–50, 50–100, 100–200, 400–500}. For MAXCUT and TSP, which involve edge weights, we train up to 200–300 nodes due to the limited computation resource. For all problems, we test on graphs of size up to 1000–1200.

During testing, instead of using Active Search as in [6], we simply use the greedy policy. This gives us much faster inference, while still being powerful enough. We modify existing open-source code to implement both S2V-DQN [2] and PN-AC [3]. Our code is publicly available [4].

## 5.1 Comparison of solution quality

To evaluate the solution quality on test instances, we use the *approximation ratio* of each method relative to the optimal solution, averaged over the set of test instances. The approximation ratio of a solution $S$ to a problem instance $G$ is defined as $\mathcal{R}(S, G) = \max(\frac{OPT(G)}{c(h(S))}, \frac{c(h(S))}{OPT(G)})$, where $c(h(S))$ is the objective value of solution $S$, and $OPT(G)$ is the best-known solution value of instance $G$.

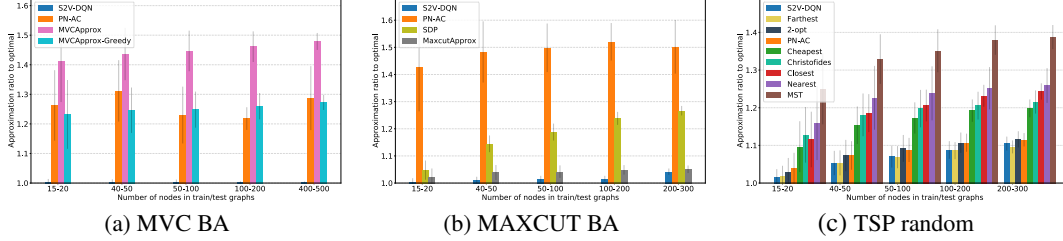

| (a) MVC BA | (b) MAXCUT BA | (c) TSP random |

Figure 2: Approximation ratio on 1000 test graphs. Note that on MVC, our performance is pretty close to optimal. In this figure, training and testing graphs are generated according to the same distribution.

Figure 2 shows the average approximation ratio across the three problems; other graph types are in Figure D.1 in the appendix. In all of these figures, a lower approximation ratio is better. Overall, our proposed method, S2V-DQN, performs significantly better than other methods. In MVC, the performance of S2V-DQN is particularly good, as the approximation ratio is roughly 1 and the bar is barely visible.

The PN-AC algorithm performs well on TSP, as expected. Since the TSP graph is essentially fully-connected, graph structure is not as important. On problems such as MVC and MAXCUT, where graph information is more crucial, our algorithm performs significantly better than PN-AC. For TSP, The Farthest and 2-opt algorithm perform as well as S2V-DQN, and slightly better in some cases. However, we will show later that in real-world TSP data, our algorithm still performs better.

## 5.2 Generalization to larger instances

The graph embedding framework enables us to train and test on graphs of different sizes, since the same set of model parameters are used. How does the performance of the learned algorithm using small graphs generalize to test graphs of larger sizes? To investigate this, we train S2V-DQN on graphs with 50–100 nodes, and test its generalization ability on graphs with up to 1200 nodes. Table 2 summarizes the results, and full results are in Appendix D.3.

Table 2: S2V-DQN's generalization ability. Values are average approximation ratios over 1000 test instances. These test results are produced by S2V-DQN algorithms trained on graphs with 50-100 nodes.

| Test Size | 50-100 | 100-200 | 200-300 | 300-400 | 400-500 | 500-600 | 1000-1200 |
|---|---|---|---|---|---|---|---|
| MVC (BA) | 1.0033 | 1.0041 | 1.0045 | 1.0040 | 1.0045 | 1.0048 | 1.0062 |
| MAXCUT (BA) | 1.0150 | 1.0181 | 1.0202 | 1.0188 | 1.0123 | 1.0177 | 1.0038 |
| TSP (clustered) | 1.0730 | 1.0895 | 1.0869 | 1.0918 | 1.0944 | 1.0975 | 1.1065 |

We can see that S2V-DQN achieves a very good approximation ratio. Note that the "optimal" value used in the computation of approximation ratios may not be truly optimal (due to the solver time cutoff at 1 hour), and so CPLEX's solutions do typically get worse as problem size grows. This is why sometimes we can even get better approximation ratio on larger graphs.

## 5.3 Scalability & Trade-off between running time and approximation ratio

To construct a solution on a test graph, our algorithm has polynomial complexity of $O(k|E|)$ where $k$ is number of greedy steps (at most the number of nodes $|V|$) and $|E|$ is number of edges. For instance, on graphs with 1200 nodes, we can find the solution of MVC within 11 seconds using a single GPU, while getting an approximation ratio of $1.0062$. For dense graphs, we can also sample the edges for the graph embedding computation to save time, a measure we will investigate in the future.

Figure 3 illustrates the approximation ratios of various approaches as a function of running time. All algorithms report a single solution at termination, whereas CPLEX reports multiple improving solutions, for which we recorded the corresponding running time and approximation ratio. Figure D.3 (Appendix D.7) includes other graph sizes and types, where the results are consistent with Figure 3.

Figure 3 shows that, for MVC, we are slightly slower than the approximation algorithms but enjoy a much better approximation ratio. Also note that although CPLEX found the first feasible solution quickly, it also has much worse ratio; the second improved solution found by CPLEX takes similar or longer time than our S2V-DQN, but is still of worse quality. For MAXCUT, the observations are still consistent. One should be aware that sometimes our algorithm can obtain better results than 1-hour CPLEX, which gives ratios below 1.0. Furthermore, sometimes S2V-DQN is even faster than the

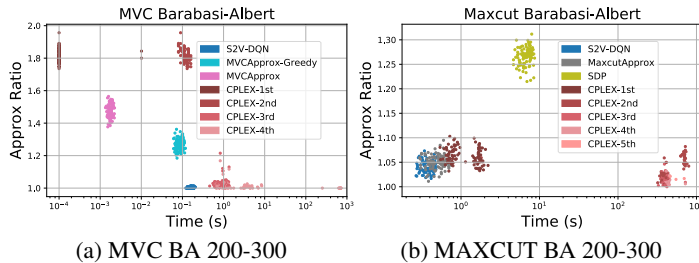

Figure 3: Time-approximation trade-off for MVC and MAX-CUT. In this figure, each dot represents a solution found for a single problem instance, for 100 instances. For CPLEX, we also record the time and quality of each solution it finds, e.g. CPLEX-1st means the first feasible solution found by CPLEX.

| (a) MVC BA 200-300 | (b) MAXCUT BA 200-300 |

*MaxcutApprox*, although this comparison is not exactly fair, since we use GPUs; however, we can still see that our algorithm is efficient.

## 5.4 Experiments on real-world datasets

In addition to the experiments for synthetic data, we identified sets of publicly available benchmark or real-world instances for each problem, and performed experiments on them. A summary of results is in Table 3, and details are given in Appendix C. S2V-DQN significantly outperforms all competing methods for MVC, MAXCUT and TSP.

Table 3: Realistic data experiments, results summary. Values are average approximation ratios.

| Problem | Dataset | S2V-DQN | Best Competitor | 2nd Best Competitor |
|---------|---------|---------|-----------------|---------------------|
| MVC | MemeTracker | **1.0021** | 1.2220 (MVCApprox-Greedy) | 1.4080 (MVCApprox) |
| MAXCUT | Physics | **1.0223** | 1.2825 (MaxcutApprox) | 1.8996 (SDP) |
| TSP | TSPLIB | **1.0475** | 1.0800 (Farthest) | 1.0947 (2-opt) |

## 5.5 Discovery of interesting new algorithms

We further examined the algorithms learned by S2V-DQN, and tried to interpret what greedy heuristics have been learned. We found that S2V-DQN is able to discover new and interesting algorithms which intuitively make sense but have not been analyzed before. For instance, S2V-DQN discovers an algorithm for MVC where nodes are selected to balance between their degrees and the connectivity of the remaining graph (Appendix Figures D.4 and D.7). For MAXCUT, S2V-DQN discovers an algorithm where nodes are picked to avoid cancelling out existing edges in the cut set (Appendix Figure D.5). These results suggest that S2V-DQN may also be a good assistive tool for discovering new algorithms, especially in cases when the graph optimization problems are new and less well-studied.

## 6 Conclusions

We presented an end-to-end machine learning framework for automatically designing greedy heuristics for hard combinatorial optimization problems on graphs. Central to our approach is the combination of a deep graph embedding with reinforcement learning. Through extensive experimental evaluation, we demonstrate the effectiveness of the proposed framework in learning greedy heuristics as compared to manually-designed greedy algorithms. The excellent performance of the learned heuristics is consistent across multiple different problems, graph types, and graph sizes, suggesting that the framework is a promising new tool for designing algorithms for graph problems.

**Acknowledgments**

This project was supported in part by NSF IIS-1218749, NIH BIGDATA 1R01GM108341, NSF CAREER IIS-1350983, NSF IIS-1639792 EAGER, NSF CNS-1704701, ONR N00014-15-1-2340, Intel ISTC, NVIDIA and Amazon AWS. Dilkina is supported by NSF grant CCF-1522054 and ExxonMobil.

## Footnotes

[2] https://github.com/Hanjun-Dai/graphnn

[3] https://github.com/devsisters/pointer-network-tensorflow

[4] https://github.com/Hanjun-Dai/graph_comb_opt

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
