[Supplementary Material · nips_2017-appendix.pdf]

# Appendix

## A Related Work

**Machine learning for combinatorial optimization.** Reinforcement learning is used to solve a job-shop flow scheduling problem in [38]. Boyan and Moore [7] use regression to learn good restart rules for local search algorithms. Both of these methods require hand-designed, problem-specific features, a limitation with the learned graph embedding.

**Machine learning for branch-and-bound.** *Learning to search* in branch-and-bound is another related research thread. This thread includes machine learning methods for branching [26, 22], tree node selection [16, 34], and heuristic selection [35, 23]. In comparison, our work promotes an even tighter integration of learning and optimization.

**Deep learning for continuous optimization.** In continuous optimization, methods have been proposed for learning an update rule for gradient descent [2, 27] and solving black-box optimization problems [8]; these are very interesting ideas that highlight the possibilities for better algorithm design through learning.

## B Set Covering Problem

We also applied our framework to the classical Set Covering Problem (SCP). SCP is interesting because it is not a graph problem, but can be formulated as one. Our framework is capable of addressing such problems seamlessly, as we will show in the coming sections of the appendix which detail the performance of S2V-DQN as compared to other methods.

**Set Covering Problem (SCP)**: Given a bipartite graph $G$ with node set $V := \mathcal{U} \cup \mathcal{C}$, find a subset of nodes $S \subseteq \mathcal{C}$ such that every node in $\mathcal{U}$ is covered, i.e. $u \in \mathcal{U} \Leftrightarrow \exists s \in S$ s.t. $(u, s) \in E$, and $|S|$ is minimized. Note that an edge $(u, s), u \in \mathcal{U}, s \in \mathcal{C}$, exists whenever subset $s$ includes element $u$.

**Meta-algorithm:** Same as MVC; the termination criterion checks whether all nodes in $\mathcal{U}$ have been covered.

**RL formulation:** In SCP, the state is a function of the subset of nodes of $\mathcal{C}$ selected so far; an action is to add node of $\mathcal{C}$ to the partial solution; the reward is -1; the termination criterion is met when all nodes of $\mathcal{U}$ are covered; no helper function is needed.

**Baselines for SCP:** We include *Greedy*, which iteratively selects the node of $\mathcal{C}$ that is not in the current partial solution and that has the most uncovered neighbors in $\mathcal{U}$ [25]. We also used *LP*, another $O(\log|\mathcal{U}|)$-approximation that solves a linear programming relaxation of SCP, and rounds the resulting fractional solution in decreasing order of variable values (SortLP-1 in [31]).

## C Experimental Results on Realistic Data

In this section, we show results on realistic nstances for all four problems. In particular, for MVC and SCP, we used the MemeTracker graph to formulate network diffusion optimization problems. For MAXCUT and TSP, we used benchmark instances that arise in physics and transportation, respectively.

### C.1 Minimum Vertex Cover

As mentioned in the introduction, the MVC problem is related to the efficient spreading of information in networks, where one wants to cover as few nodes as possible such that all nodes have at least one neighbor in the cover. The MemeTracker graph [5] is a network of who-copies-whom, where nodes represent news sites or blogs, and a (directed) edge from $u$ to $v$ means that $v$ frequently copies phrases (or memes) from $u$. The network is learned from real traces in [13], having 960 nodes and 5000 edges. The dataset also provides the average transmission time $\Delta_{u,v}$ between a pair of nodes, i.e. how much later $v$ copies $u$'s phrases after their publication online, on average. As done in [21],

we use these average transmission times to compute a diffusion probability $P(u, v)$ on the edge, such that $P(u, v) = \alpha \cdot \dfrac{1}{\Delta_{u,v}}$, where $\alpha$ is a parameter of the diffusion model. In both MVC and SCP, we use $\alpha = 0.1$, but results are consistent for other values we have considered. For pairs of nodes that have edges in both directions, i.e. $(u, v)$ and $(v, u)$, we take the average probability to obtain an undirected version of the graph, as MVC is defined for undirected graphs.

Following the widely-adopted Independent Cascade model (see [10] for example), we sample a diffusion cascade from the full graph by independently keeping an edge with probability $P(u, v)$. We then consider the largest connected component in the graph as a single training instance, and train S2V-DQN on a set of such sampled diffusion graphs. The aim is to test the learned model on the (undirected version of the) *full* MemeTracker graph.

Experimentally, an optimal cover has 473 nodes, whereas S2V-DQN finds a cover with 474 nodes, only one more than the optimum, at an approximation ratio of 1.002. In comparison, MVCApprox and MVCApprox-Greedy find much larger covers with 666 and 578 nodes, at approximation ratios of 1.408 and 1.222, respectively.

## C.2  Maximum Cut

A library of Maximum Cut instances is publicly available [6], and includes synthetic and realistic instances that are widely used in the optimization community (see references at library website). We perform experiments on a subset of the instances available, namely ten problems from Ising spin glass models in physics, given that they are realistic and manageable in size (the first 10 instances in Set2 of the library). All ten instances have 125 nodes and 375 edges, with edge weights in $\{-1, 0, 1\}$.

To train our S2V-DQN model, we constructed a training dataset by perturbing the instances, adding random Gaussian noise with mean 0 and standard deviation 0.01 to the edge weights. After training, the learned model is used to construct a cut-set greedily on each of the ten instances, as before.

Table C.1 shows that S2V-DQN finds near-optimal solutions (optimal in 3/10 instances) that are much better than those found by competing methods.

Table C.1: MAXCUT results on the ten instances described in C.2; values reported are cut weights of the solution returned by each method, where larger values are better (best in bold). Bottom row is the average approximation ratio (lower is better).

| Instance | OPT | S2V-DQN | MaxcutApprox | SDP |
|----------|-----|---------|--------------|-----|
| G54100 | 110 | **108** | 80 | 54 |
| G54200 | 112 | **108** | 90 | 58 |
| G54300 | 106 | **104** | 86 | 60 |
| G54400 | 114 | **108** | 96 | 56 |
| G54500 | 112 | **112** | 94 | 56 |
| G54600 | 110 | **110** | 88 | 66 |
| G54700 | 112 | **108** | 88 | 60 |
| G54800 | 108 | **108** | 76 | 54 |
| G54900 | 110 | **108** | 88 | 68 |
| G5410000 | 112 | **108** | 80 | 54 |
| Approx. ratio | 1 | **1.02** | 1.28 | 1.90 |

## C.3  Traveling Salesman Problem

We use the standard TSPLIB library [32] which is publicly available [7]. We target 38 TSPLIB instances with sizes ranging from 51 to 318 cities (or nodes). We do not tackle larger instances as we are limited by the memory of a single graphics card. Nevertheless, most of the instances addressed here are larger than the largest instance used in [6].

We apply S2V-DQN in "Active Search" mode, similarly to [6]: no upfront training phase is required, and the reinforcement learning algorithm 1 is applied on-the-fly on each instance. The best tour encountered over the episodes of the RL algorithm is stored.

Table C.2 shows the results of our method and six other TSP algorithms. On all but 6 instances, S2V-DQN finds the best tour among all methods. The average approximation ratio of S2V-DQN is also the smallest at 1.05.

Table C.2: TSPLIB results: Instances are sorted by increasing size, with the number at the end of an instance's name indicating its size. Values reported are the cost of the tour found by each method (lower is better, best in bold). Bottom row is the average approximation ratio (lower is better).

| Instance | OPT | S2V-DQN | Farthest | 2-opt | Cheapest | Christofides | Closest | Nearest | MST |
|---|---|---|---|---|---|---|---|---|---|
| eil51 | 426 | **439** | 467 | 446 | 494 | 527 | 488 | 511 | 614 |
| berlin52 | 7,542 | **7,542** | 8,307 | 7,788 | 9,013 | 8,822 | 9,004 | 8,980 | 10,402 |
| st70 | 675 | **696** | 712 | 753 | 776 | 836 | 814 | 801 | 858 |
| eil76 | 538 | **564** | 583 | 591 | 607 | 646 | 615 | 705 | 743 |
| pr76 | 108,159 | **108,446** | 119,692 | 115,460 | 125,935 | 137,258 | 128,381 | 153,462 | 133,471 |
| rat99 | 1,211 | **1,280** | 1,314 | 1,390 | 1,473 | 1,399 | 1,465 | 1,558 | 1,665 |
| kroA100 | 21,282 | **21,897** | 23,356 | 22,876 | 24,309 | 26,578 | 25,787 | 26,854 | 30,516 |
| kroB100 | 22,141 | **22,692** | 23,222 | 23,496 | 25,582 | 25,714 | 26,875 | 29,158 | 28,807 |
| kroC100 | 20,749 | **21,074** | 21,699 | 23,445 | 25,264 | 24,582 | 25,640 | 26,327 | 27,636 |
| kroD100 | 21,294 | 22,102 | **22,034** | 23,967 | 25,204 | 27,863 | 25,213 | 26,947 | 28,599 |
| kroE100 | 22,068 | 22,913 | 23,516 | **22,800** | 25,900 | 27,452 | 27,313 | 27,585 | 30,979 |
| rd100 | 7,910 | **8,159** | 8,944 | 8,757 | 8,980 | 10,002 | 9,485 | 9,938 | 10,467 |
| eil101 | 629 | **659** | 673 | 702 | 693 | 728 | 720 | 817 | 847 |
| lin105 | 14,379 | **15,023** | 15,193 | 15,536 | 16,930 | 16,568 | 18,592 | 20,356 | 21,167 |
| pr107 | 44,303 | **45,113** | 45,905 | 47,058 | 52,816 | 49,192 | 52,765 | 48,521 | 55,956 |
| pr124 | 59,030 | **61,623** | 65,945 | 64,765 | 65,316 | 64,591 | 68,178 | 69,297 | 82,761 |
| bier127 | 118,282 | **121,576** | 129,495 | 128,103 | 141,354 | 135,134 | 145,516 | 129,333 | 153,658 |
| ch130 | 6,110 | **6,270** | 6,498 | 6,470 | 7,279 | 7,367 | 7,434 | 7,578 | 8,280 |
| pr136 | 96,772 | **99,474** | 105,361 | 110,531 | 109,586 | 116,069 | 105,778 | 120,769 | 142,438 |
| pr144 | 58,537 | **59,436** | 61,974 | 60,321 | 73,032 | 74,684 | 73,613 | 61,652 | 77,704 |
| ch150 | 6,528 | **6,985** | 7,210 | 7,232 | 7,995 | 7,641 | 7,914 | 8,191 | 9,203 |
| kroA150 | 26,524 | **27,888** | 28,658 | 29,666 | 29,963 | 32,631 | 31,341 | 33,612 | 38,763 |
| kroB150 | 26,130 | **27,209** | 27,404 | 29,517 | 31,589 | 33,260 | 31,616 | 32,825 | 35,289 |
| pr152 | 73,682 | **75,283** | 75,396 | 77,206 | 88,531 | 82,118 | 86,915 | 85,699 | 90,292 |
| u159 | 42,080 | **45,433** | 46,789 | 47,664 | 49,986 | 48,908 | 52,009 | 53,641 | 54,399 |
| rat195 | 2,323 | **2,581** | 2,609 | 2,605 | 2,806 | 2,906 | 2,935 | 2,753 | 3,163 |
| d198 | 15,780 | 16,453 | **16,138** | 16,596 | 17,632 | 19,002 | 17,975 | 18,805 | 19,339 |
| kroA200 | 29,368 | **30,965** | 31,949 | 32,760 | 35,340 | 37,487 | 36,025 | 35,794 | 40,234 |
| kroB200 | 29,437 | 31,692 | **31,522** | 33,107 | 35,412 | 34,490 | 36,532 | 36,976 | 40,615 |
| ts225 | 126,643 | **136,302** | 140,626 | 138,101 | 160,014 | 145,283 | 151,887 | 152,493 | 188,008 |
| tsp225 | 3,916 | **4,154** | 4,280 | 4,278 | 4,470 | 4,733 | 4,780 | 4,749 | 5,344 |
| pr226 | 80,369 | **81,873** | 84,130 | 89,262 | 91,023 | 98,101 | 100,118 | 94,389 | 114,373 |
| gil262 | 2,378 | **2,537** | 2,623 | 2,597 | 2,800 | 2,963 | 2,908 | 3,211 | 3,336 |
| pr264 | 49,135 | **52,364** | 54,462 | 54,547 | 57,602 | 55,955 | 65,819 | 58,635 | 66,400 |
| a280 | 2,579 | **2,867** | 3,001 | 2,914 | 3,128 | 3,125 | 2,953 | 3,302 | 3,492 |
| pr299 | 48,191 | **51,895** | 51,903 | 54,914 | 58,127 | 58,660 | 59,740 | 61,243 | 65,617 |
| lin318 | 42,029 | 45,375 | 45,918 | **45,263** | 49,440 | 51,484 | 52,353 | 54,019 | 60,939 |
| linhp318 | 41,345 | 45,444 | 45,918 | **45,263** | 49,440 | 51,484 | 52,353 | 54,019 | 60,939 |
| Approx. ratio | 1 | **1.05** | 1.08 | 1.09 | 1.18 | 1.2 | 1.21 | 1.24 | 1.37 |

## C.4 Set Covering Problem

The SCP is also related to the diffusion optimization problem on graphs; for instance, the proof of hardness in the classical [20] paper uses SCP for the reduction. As in MVC, we leverage the MemeTracker graph, albeit differently.

We use the same cascade model as in MVC to assign the edge probabilities, and sample graphs from it in the same way. Let $\mathcal{R}^G(u)$ be the set of nodes reachable from $u$ in a sampled graph $\tilde{G}$. For every node $u$ in $G$, there are two corresponding nodes in the SCP instance, $u_{\mathcal{C}} \in \mathcal{C}$ and $u_{\mathcal{U}} \in \mathcal{U}$. An edge exists between $u_{\mathcal{C}} \in \mathcal{C}$ and $v_{\mathcal{U}} \in \mathcal{U}$ if and only if $v \in \mathcal{R}^G(u)$. In other words, each node in the sampled graph $G$ has a set consisting of the other nodes that it can reach in $G$. As such, the SCP reduces to finding the smallest set of nodes whose union can reach all other nodes. We generate training and testing graphs according to this same process, with $\alpha = 0.1$.

Experimentally, we test S2V-DQN and the other baseline algorithms on a set of 1000 test graphs. S2V-DQN achieves an average approximation ratio of 1.001, only slightly behind LP, which achieves 1.0009, and well ahead of Greedy at 1.03.

# D   Experiment Details

## D.1   Problem instance generation

### D.1.1   Minimum Vertex Cover

For the Minimum Vertex Cover (MVC) problem, we generate random Erdős-Renyi (edge probability 0.15) and Barabasi-Albert (average degree 4) graphs of various sizes, and use the integer programming solver CPLEX 12.6.1 with a time cutoff of 1 hour to compute optimal solutions for the generated instances. When CPLEX fails to find an optimal solution, we report the best one found within the time cutoff as "optimal". All graphs were generated using the NetworkX [8] package in Python.

### D.1.2   Maximum Cut

For the Maximum Cut (MAXCUT) problem, we use the same graph generation process as in MVC, and augment each edge with a weight drawn uniformly at random from $[0, 1]$. We use a quadratic formulation of MAXCUT with CPLEX 12.6.1. and a time cutoff of 1 hour to compute optimal solutions, and report the best solution found as "optimal".

### D.1.3   Traveling Salesman Problem

For the (symmetric) 2-dimensional TSP, we use the instance generator of the 8th DIMACS Implementation Challenge [9] [18] to generate two types of Euclidean instances: "random" instances consist of $n$ points scattered uniformly at random in the $[10^6, 10^6]$ square, while "clustered" instances consist of $n$ points that are clustered into $n/100$ clusters; generator details are described in page 373 of [18].

To compute optimal TSP solutions for both TSP, we use the state-of-the-art solver, Concorde [10] [3], with a time cutoff of 1 hour.

### D.1.4   Set Covering Problem

For the SCP, given a number of node $n$, roughly $0.2n$ nodes are in node-set $\mathcal{C}$, and the rest in node-set $\mathcal{U}$. An edge between nodes in $\mathcal{C}$ and $\mathcal{U}$ exists with probability either 0.05 or 0.1, which can be seen as "density" values, and commonly appear for instances used in optimization papers on SCP [5]. We guarantee that each node in $\mathcal{U}$ has at least 2 edges, and each node in $\mathcal{C}$ has at least one edge, a standard measure for SCP instances [5]. We also use CPLEX 12.6.1. with a time cutoff of 1 hour to compute a near-optimal or optimal solution to a SCP instance.

## D.2   Full results on solution quality

Table D.1 is a complete version of Table 2 that appears in the main text.

## D.3   Full results on generalization

The full generalization results can be found in Table D.1, D.2, D.3, D.4, D.5, D.6, D.7 and D.8.

## D.4   Experiment Configuration of S2V-DQN

The node/edge representations and hyperparameters used in our experiments is shown in Table D.9. For our method, we simply tune the hyperparameters on small graphs (i.e., the graphs with less than 50 nodes), and fix them for larger graphs.

| Train \ Test | 15-20 | 40-50 | 50-100 | 100-200 | 200-300 | 300-400 | 400-500 | 500-600 | 1000-1200 |
|---|---|---|---|---|---|---|---|---|---|
| 15-20 | 1.0032 | 1.0883 | 1.0941 | 1.0710 | 1.0484 | 1.0365 | 1.0276 | 1.0246 | 1.0111 |
| 40-50 | ＼ | 1.0037 | 1.0076 | 1.1013 | 1.0991 | 1.0800 | 1.0651 | 1.0573 | 1.0299 |
| 50-100 | ＼ | ＼ | 1.0079 | 1.0304 | 1.0570 | 1.0532 | 1.0463 | 1.0427 | 1.0238 |
| 100-200 | ＼ | ＼ | ＼ | 1.0102 | 1.0095 | 1.0136 | 1.0142 | 1.0125 | 1.0103 |
| 400-500 | ＼ | ＼ | ＼ | ＼ | ＼ | ＼ | 1.0021 | 1.0027 | 1.0057 |

Table D.1: S2V-DQN's generalization on MVC problem in ER graphs.

| Train \ Test | 15-20 | 40-50 | 50-100 | 100-200 | 200-300 | 300-400 | 400-500 | 500-600 | 1000-1200 |
|---|---|---|---|---|---|---|---|---|---|
| 15-20 | 1.0016 | 1.0027 | 1.0039 | 1.0066 | 1.0093 | 1.0106 | 1.0125 | 1.0150 | 1.0491 |
| 40-50 | ＼ | 1.0027 | 1.0051 | 1.0092 | 1.0130 | 1.0144 | 1.0161 | 1.0170 | 1.0228 |
| 50-100 | ＼ | ＼ | 1.0033 | 1.0041 | 1.0045 | 1.0040 | 1.0045 | 1.0048 | 1.0062 |
| 100-200 | ＼ | ＼ | ＼ | 1.0016 | 1.0020 | 1.0019 | 1.0021 | 1.0026 | 1.0060 |
| 400-500 | ＼ | ＼ | ＼ | ＼ | ＼ | ＼ | 1.0025 | 1.0026 | 1.0030 |

Table D.2: S2V-DQN's generalization on MVC problem in BA graphs.

| Train \ Test | 15-20 | 40-50 | 50-100 | 100-200 | 200-300 | 300-400 | 400-500 | 500-600 | 1000-1200 |
|---|---|---|---|---|---|---|---|---|---|
| 15-20 | 1.0034 | 1.0167 | 1.0407 | 1.0667 | 1.1067 | 1.1489 | 1.1885 | 1.2150 | 1.1488 |
| 40-50 | ＼ | 1.0127 | 1.0154 | 1.0089 | 1.0198 | 1.0383 | 1.0388 | 1.0384 | 1.0534 |
| 50-100 | ＼ | ＼ | 1.0112 | 1.0024 | 1.0109 | 1.0467 | 1.0926 | 1.1426 | 1.1297 |
| 100-200 | ＼ | ＼ | ＼ | 1.0005 | 1.0021 | 1.0211 | 1.0373 | 1.0612 | 1.2021 |
| 200-300 | ＼ | ＼ | ＼ | ＼ | 1.0106 | 1.0272 | 1.0487 | 1.0700 | 1.1759 |

Table D.3: S2V-DQN's generalization on MAXCUT problem in ER graphs.

| Train \ Test | 15-20 | 40-50 | 50-100 | 100-200 | 200-300 | 300-400 | 400-500 | 500-600 | 1000-1200 |
|---|---|---|---|---|---|---|---|---|---|
| 15-20 | 1.0055 | 1.0119 | 1.0176 | 1.0276 | 1.0357 | 1.0386 | 1.0335 | 1.0411 | 1.0331 |
| 40-50 | ＼ | 1.0107 | 1.0119 | 1.0139 | 1.0144 | 1.0119 | 1.0039 | 1.0085 | 0.9905 |
| 50-100 | ＼ | ＼ | 1.0150 | 1.0181 | 1.0202 | 1.0188 | 1.0123 | 1.0177 | 1.0038 |
| 100-200 | ＼ | ＼ | ＼ | 1.0166 | 1.0183 | 1.0166 | 1.0104 | 1.0166 | 1.0156 |
| 200-300 | ＼ | ＼ | ＼ | ＼ | 1.0420 | 1.0394 | 1.0290 | 1.0319 | 1.0244 |

Table D.4: S2V-DQN's generalization on MAXCUT problem in BA graphs.

| Train \ Test | 15-20 | 40-50 | 50-100 | 100-200 | 200-300 | 300-400 | 400-500 | 500-600 | 1000-1200 |
|---|---|---|---|---|---|---|---|---|---|
| 15-20 | 1.0147 | 1.0511 | 1.0702 | 1.0913 | 1.1022 | 1.1102 | 1.1124 | 1.1156 | 1.1212 |
| 40-50 | ＼ | 1.0533 | 1.0701 | 1.0890 | 1.0978 | 1.1051 | 1.1583 | 1.1587 | 1.1609 |
| 50-100 | ＼ | ＼ | 1.0701 | 1.0871 | 1.0983 | 1.1034 | 1.1071 | 1.1101 | 1.1171 |
| 100-200 | ＼ | ＼ | ＼ | 1.0879 | 1.0980 | 1.1024 | 1.1056 | 1.1080 | 1.1142 |
| 200-300 | ＼ | ＼ | ＼ | ＼ | 1.1049 | 1.1090 | 1.1084 | 1.1114 | 1.1179 |

Table D.5: S2V-DQN's generalization on TSP in random graphs.

| Train \ Test | 15-20 | 40-50 | 50-100 | 100-200 | 200-300 | 300-400 | 400-500 | 500-600 | 1000-1200 |
|---|---|---|---|---|---|---|---|---|---|
| 15-20 | 1.0214 | 1.0591 | 1.0761 | 1.0958 | 1.0938 | 1.0966 | 1.1009 | 1.1012 | 1.1085 |
| 40-50 | ＼ | 1.0564 | 1.0740 | 1.0939 | 1.0904 | 1.0951 | 1.0974 | 1.1014 | 1.1091 |
| 50-100 | ＼ | ＼ | 1.0730 | 1.0895 | 1.0869 | 1.0918 | 1.0944 | 1.0975 | 1.1065 |
| 100-200 | ＼ | ＼ | ＼ | 1.1009 | 1.0979 | 1.1013 | 1.1059 | 1.1048 | 1.1091 |
| 200-300 | ＼ | ＼ | ＼ | ＼ | 1.1012 | 1.1049 | 1.1080 | 1.1067 | 1.1112 |

Table D.6: S2V-DQN's generalization on TSP in clustered graphs.

Figure D.1: Approximation ratio on 1000 test graphs. Note that on MVC, our performance is pretty close to optimal. In this figure, training and testing graphs are generated according to the same distribution.

## D.5    Stabilizing the training of S2V-DQN

For the learning rate, we use exponential decay after a certain number of steps, where the decay factor is fixed to 0.95. We also anneal the exploration probability $\epsilon$ from 1.0 to 0.05 in a linear way. For the discounting factor used in MDP, we use 1.0 for MVC, MAXCUT and SCP. For TSP, we use 0.1.

We also normalize the intermediate reward by the maximum number of nodes. For Q-learning, it is also important to disentangle the actual $Q$ with obsolete $\tilde{Q}$, as mentioned in [29].

Also for TSP with insertion helper function, we find it works better with *negative* version of designed reward function. This sounds counter intuitive at the beginning. However, since typically the RL

| Train \ Test | 15-20 | 40-50 | 50-100 | 100-200 | 200-300 | 300-400 | 400-500 | 500-600 | 1000-1200 |
|---|---|---|---|---|---|---|---|---|---|
| 15-20 | 1.0055 | 1.0170 | 1.0436 | 1.1757 | 1.3910 | 1.6255 | 1.8768 | 2.1339 | 3.0574 |
| 40-50 | ↘ | 1.0039 | 1.0083 | 1.0241 | 1.0452 | 1.0647 | 1.0792 | 1.0858 | 1.0775 |
| 50-100 | ↘ | ↘ | 1.0056 | 1.0199 | 1.0382 | 1.0614 | 1.0845 | 1.0821 | 1.0620 |
| 100-200 | ↘ | ↘ | ↘ | 1.0147 | 1.0270 | 1.0417 | 1.0588 | 1.0774 | 1.0509 |
| 200-300 | ↘ | ↘ | ↘ | ↘ | 1.0273 | 1.0415 | 1.0828 | 1.1357 | 1.2349 |

Table D.7: S2V-DQN's generalization on SCP with edge probability 0.05.

| Train \ Test | 15-20 | 40-50 | 50-100 | 100-200 | 200-300 | 300-400 | 400-500 | 500-600 | 1000-1200 |
|---|---|---|---|---|---|---|---|---|---|
| 15-20 | 1.0015 | 1.0200 | 1.0369 | 1.0795 | 1.1147 | 1.1290 | 1.1325 | 1.1255 | 1.0805 |
| 40-50 | ↘ | 1.0048 | 1.0137 | 1.0453 | 1.0849 | 1.1055 | 1.1052 | 1.0958 | 1.0618 |
| 50-100 | ↘ | ↘ | 1.0090 | 1.0294 | 1.0771 | 1.1180 | 1.1456 | 1.2161 | 1.0946 |
| 100-200 | ↘ | ↘ | ↘ | 1.0231 | 1.0394 | 1.0564 | 1.0702 | 1.0747 | 2.5055 |
| 200-300 | ↘ | ↘ | ↘ | ↘ | 1.0378 | 1.0517 | 1.0592 | 1.0556 | 1.3192 |

Table D.8: S2V-DQN's generalization on SCP with edge probability 0.1.

agent will bias towards most recent rewards, flipping the sign of reward function suggests a focus over future rewards. This is especially useful with the insertion construction. But it shows that designing a good reward function is still challenging for learning combinatorial algorithm, which we will investigate in our future work.

### D.6 Convergence of S2V-DQN

In Figure D.2, we plot our algorithm's convergence with respect to the held-out validation performance. We first obtain the convergence curve for each type of problem under every graph distribution. To visualize the convergence at the same scale, we plot the approximate ratio.

Figure D.2 shows that our algorithm converges nicely on the MVC, MAXCUT and SCP problems. For the MVC, we use the model trained on small graphs to initialize the model for training on larger ones. Since our model also generalizes well to problems with different sizes, the curve looks almost flat. For TSP, where the graph is essentially fully connected, it is harder to learn a good model based on graph structure. Nevertheless, as shown in previous section, the graph embedding can still learn good feature representations with multiple embedding iterations.

### D.7 Complete time v/s approximation ratio plots

Figure D.3 is a superset of Figure 3, including both graph types and three graph size ranges for MVC, MAXCUT and SCP.

### D.8 Additional analysis of the trade-off between time and approx. ratio

Tables D.10 and D.11 offer another perspective on the trade-off between the running time of a heuristic and the quality of the solution it finds. We ran CPLEX for MVC and MAXCUT for 10 minutes on the 200-300 node graphs, and recorded the time and value of all the solutions found by CPLEX within the limit; results shown next carry over to smaller graphs. Then, for a given method M that terminates in T seconds on a graph G and returns a solution with approximation ratio R, we asked the following 2 questions:

| Problem | Node tag | Edge feature | Embedding size $p$ | $T$ | Batch size | n-step |
|---|---|---|---|---|---|---|
| Minimum Vertex Cover | 0/1 tag | N/A | 64 | 5 | 128 | 5 |
| Maximum Cut | 0/1 tag | edge length; end node tag | 64 | 3 | 64 | 1 |
| Traveling Salesman Problem | coordinates; 0/1 tag; start/end node | edge length; end node tag | 64 | 4 | 64 | 1 |
| Set Covering Problem | 0/1 tag | N/A | 64 | 5 | 64 | 2 |

Table D.9: S2V-DQN's configuration used in Experiment.

Figure D.2: S2V-DQN convergence measured by the held-out validation performance.

1. If CPLEX is given the same amount of time T for G, how well can CPLEX do?

2. How long does CPLEX need to find a solution of same or better quality than the one the heuristic has found?

Figure D.3: Time-approximation trade-off for MVC, MAXCUT and SCP. In this figure, each dot represents a solution found for a single problem instance. For CPLEX, we also record the time and quality of each solution it finds. For example, CPLEX-1st means the first feasible solution found by CPLEX.

For the first question, the column "Approx. Ratio of Best Solution" in Tables D.10 and D.11 shows the following:

- MVC (Table D.10): The larger values for S2V-DQN imply that solutions we find quickly are of higher quality, as compared to the MVCApprox/Greedy baselines.

- MAXCUT (Table D.11): On most of the graphs, CPLEX cannot find any solution at all if given the same time as S2V-DQN or MaxcutApprox. SDP (solved with state-of-the-art CVX solver) is so slow that CPLEX finds solutions that are 10% better than those of SDP if given the same time as SDP (on ER graphs), which confirms that SDP is not time-efficient. One possible interpretation of the poor performance of SDP is that its theoretical guaranteed of 0.87 is *in expectation* over the solutions it can generate, and so the variance in the approximation ratios of these solutions may be very large.

For the second question, the column "Additional Time Needed" in Tables D.10 and D.11 shows the following:

- MVC (Table D.10): The larger values for S2V-DQN imply that solutions we find are harder to improve upon, as compared to the MVCApprox/Greedy baselines.

- MAXCUT (Table D.11): On ER (BA) graphs, CPLEX (10 minute-cutoff) cannot find a solution that is better than those of S2V-DQN or MaxcutApprox on many instances (e.g. the value (59) for S2V-DQN on ER graphs means that on $41 = 100 - 59$ graphs, CPLEX could not find a solution that is as good as S2V-DQN's). When we consider only those graphs for which CPLEX could find a better solution, S2V-DQN's solutions take significantly more time for CPLEX to beat, as compared to MaxcutApprox and SDP. The negative values for SDP indicate that CPLEX finds a solution better than SDP's in a shorter time.

Table D.10: Minimum Vertex Cover (100 graphs with 200-300 nodes): Trade-off between running time and approximation ratio. An "Approx. Ratio of Best Solution" value of 1.x% means that the solution found by CPLEX if given the same time as a certain heuristic (in the corresponding row) is x% worse, on average. "Additional Time Needed" in seconds is the additional amount of time needed by CPLEX to find a solution of value at least as good as the one found by a given heuristic; negative values imply that CPLEX finds such solutions faster than the heuristic does. Larger values are better for both metrics. The values in parantheses are the number of instances (out of 100) for which CPLEX finds some solution in the given time (for "Approx. Ratio of Best Solution"), or finds some solution that is at least as good as the heuristic's (for "Additional Time Needed").

| | Approx. Ratio of Best Solution | | Additional Time Needed | |
| --- | --- | --- | --- | --- |
| | ER | BA | ER | BA |
| S2V-DQN | 1.09 (100) | 1.81 (100) | 2.14 (100) | 137.42 (100) |
| MVCApprox-Greedy | 1.07 (100) | 1.44 (100) | 1.92 (100) | 0.83 (100) |
| MVCApprox | 1.03 (100) | 1.24 (98) | 2.49 (100) | 0.92 (100) |

Table D.11: Maximum Cut (100 graphs with 200-300 nodes): please refer to the caption of Table D.10.

| | Approx. Ratio of Best Solution | | Additional Time Needed | |
| --- | --- | --- | --- | --- |
| | ER | BA | ER | BA |
| S2V-DQN | N/A (0) | 1081.45 (1) | 8.99 (59) | 402.05 (34) |
| MaxcutApprox | 1.00 (48) | 340.11 (3) | -0.23 (50) | 218.19 (57) |
| SDP | 0.90 (100) | 0.84 (100) | -6.06 (100) | -5.54 (100) |

## D.9 Visualization of solutions

In Figure D.4, D.5 and D.6, we visualize solutions found by our algorithm for MVC, MAXCUT and TSP problems, respectively. For the ease of presentation, we only visualize small-size graphs. For MVC and MAXCUT, the graph is of the ER type and has 18 nodes. For TSP, we show solutions for a "random" instance (18 points) and a "clustered" one (15 points).

For MVC and MAXCUT, we show two step by step examples where S2V-DQN finds the optimal solution. For MVC, it seems we are picking the node which covers the most edges in the current state. However, in a more detailed visualization in Appendix D.10, we show that our algorithm learns a smarter greedy or dynamic programming like strategy. While picking the nodes, it also learns how to keep the connectivity of graph by scarifying the intermediate edge coverage a little bit.

In the example of MAXCUT, it is even more interesting to see that the algorithm did not pick the node which gives the largest intermediate reward at the beginning. Also in the intermediate steps, the agent seldom chooses a node which would cancel out the edges that are already in the cut set. This also shows the effectiveness of graph state representation, which provides useful information to support the agent's node selection decisions. For TSP, we visualize an optimal tour and one found by S2V-DQN for two instances. While the tours found by S2V-DQN differ slightly from the optimal solutions visualized, they are of comparable cost and look qualitatively acceptable. The cost of the tours found by S2V-DQN is within $0.07\%$ and $0.5\%$ of optimum, respectively.

Figure D.4: Minimum Vertex Cover: an optimal solution to an ER graph instance found by S2V-DQN. Selected node in each step is colored in orange, and nodes in the partial solution up to that iteration are colored in black. Newly covered edges are in thick green, previously covered edges are in red, and uncovered edges in black. We show that the agent is not only picking the node with large degree, but also trying to maintain the connectivity after removal of the covered edges. For more detailed analysis, please see Appendix D.10.

Figure D.5: Maximum Cut: an optimal solution to ER graph instance found by S2V-DQN. Nodes are partitioned into two sets: white or black nodes. At each iteration, the node selected to join the set of black nodes is highlighted in orange, and the new cut edges it produces are in green. Cut edges from previous iteration are in red (Best viewed in color). It seems the agent will try to involve the nodes that won't cancel out the edges in current cut set.

### D.10 Detailed visualization of learned MVC strategy

In Figure D.7, we show a detailed comparison with our learned strategy and two other simple heuristics. We find that the S2V-DQN can learn a much smarter strategy, where the agent is trying to maintain the connectivity of graph during node picking and edge removal.

Figure D.6: Traveling Salesman Problem. Left: optimal tour to a "random" instance with 18 points (all edges are red), compared to a tour found by our method next to it. For our tour, edges that are not in the optimal tour are shown in green. Our tour is $0.07\%$ longer than an optimal tour. Right: a "clustered" instance with 15 points; same color coding as left figure. Our tour is $0.5\%$ longer than an optimal tour. (Best viewed in color).

## D.11 Experiment Configuration of PN-AC

We implemented PN-AC to the best of our capabilities. Note that it is quite possible that there are minor differences between our implementation and Bello et al. [6] that might have resulted in performance not as good as reported in that paper.

For experiments of PN-AC across all tasks, we follow the configurations provided in [6]: *a*) For the input data, we use mini-batches of 128 sequences with 0-paddings to the maximal input length (which is the maximal number of nodes) in the training data. *b*) For node representation, we use coordinates for TSP, so the input dimension is 2. For MVC, MAXCUT and SCP, we represent nodes based on the adjacency matrix of the graph. To get a fixed dimension representation for each node, we use SVD to get a low-rank approximation of the adjacency matrix. We set the rank as 8, so that each node in the input sequence is represented by a 8-dimensional vector. *c*) For the network structure, we use standard single-layer LSTM cells with 128 hidden units for both encoder and decoder parts of the pointer networks. *d*) For the optimization method, we train the PN-AC model with the Adam optimizer [24] and use an initial learning rate of $10^{-3}$ that decay every 5000 steps by a factor of 0.96. *e*) For the glimpse trick, we exactly use one-time glimpse in our implementation, as described in the original PN-AC paper. *f*) We initialize all the model parameters uniformly random within $[-0.08, 0.08]$ and clip the $L2$ norm of the gradients to 1.0. *g*) For the baseline function in the actor-critic algorithm, we tried the critic network in our implementation, but it hurts the performance according to our experiments. So we use the exponential moving average performance of the sampled solution from the pointer network as the baseline.

**Consistency with the results from Bello et al. [6]** Though our TSP experiment setting is not exactly the same as Bello et al. [6], we still include some of the results directly here, for the sake of completeness. We applied the insertion heuristic to PN-AC as well, and all the results reported in our paper are with the insertion heuristic. We compare the approximation ratio reported by Bello et al. [6] verses which reported by our implementation. For TSP20: 1.02 vs 1.03 (reported in our paper); TSP50: 1.05 vs 1.07 (reported in our paper); TSP100: 1.07 vs 1.09 (reported in our paper). Note that we have variable graph size in each setting (where the original PN-AC is only reported on fixed graph size), which makes the task more difficult. Therefore, we think the performance gap here is pretty reasonable.

Figure D.7: Step-by-step comparison between our S2V-DQN and two greedy heuristics. We can see our algorithm will also favor the large degree nodes, but it will also do something smartly: instead of breaking the graph into several disjoint components, our algorithm will try the best to keep the graph connected.

## Footnotes

[5]http://snap.stanford.edu/netinf/#data

[6]http://www.optsicom.es/maxcut/#instances

[7]http://elib.zib.de/pub/mp-testdata/tsp/tsplib/tsp/index.html

[8]https://networkx.github.io/

[9]http://dimacs.rutgers.edu/Challenges/TSP/

[10]http://www.math.uwaterloo.ca/tsp/concorde/