[Reviews · NeurIPS 2017]

Reviewer 1



The authors propose a reinforcement learning strategy to learn new heuristic (specifically, greedy) strategies for solving graph-based combinatorial problems. An RL framework is combined with a graph embedding approach. The RL approach effectively learns a greedy policy for selecting constructing an approximate solution. The approach is innovative and the empirical results appear promising. An important advantage of the work is that the learned policy is not restricted to a fixed problem size, in contrast to earlier work. One drawback of the paper is that I found the writing unnecessarily dense and unclear at various places. In particular, it would be good to include more details on the intuitive description of the approach. I believe the ideas can be stated clearly in words because the concept of learning a greedy policy is not that different from learning any policy as done in RL (Learning the “next move to make” in a game is quite analogous to learning what is the next node in the graph to select. So, the authors can focus more on what makes this work different from learning a game strategy.) The over reliance on formal notation does not help. I did not fully check the formal details but some points were unclear. For example, a state is defined as a sequence of action nodes on the graph. Each node in the tagged graph is represented by its p-dimensional embedding (a p-dimensional vector). A state is then defined as the sum of the vectors corresponding to the set of action nodes so far. Why is this choice of state definition the “right one”? The empirical results are promising but some further insights would be helpful. For most combinatorial problems there is a known basic greedy strategy that already performs quite well. Does RL rediscover those strategies for the domains under consideration? (If the framework has the potential to discover new algorithmic strategy, it would be good to know that the approach can also re-discover known strategies.)

Reviewer 2



The authors aim to solve problems in combinatorial optimization by exploiting large datasets of solved problem instances to learn a solution algorithm. They focus on problems that can be expressed as graphs, which is a very general class. Their approach is to train a greedy algorithm to build up solutions by reinforcement learning (RL). The algorithm uses a learned evaluation function, Q, which maps [a partial solution] and [a candidate change to the solution] to [a value]. To embed graphs of different shapes and sizes in a fixed-length format, they use a learned "structure2vec" function (introduced in previous work). The structure2vec function is applied to a partial solution, which is comprised of the input graph where each node is augmented with a per-node feature that indicates a possible solution to the problem. For instance, in minimum vertex cover, the extra feature indicates whether the node is in the cover set. The structure2vec and Q functions are implemented as neural networks, and several variants of Q learning are applied to train them. They test on 3 tasks: minimum vertex cover, maximum cut, and traveling salesman problem. They compare their learned model's performance to Pointer Networks, as well as a variety of non-learned algorithms. They show that their S2V-DQN algorithm has much better performance than all competitors in most cases, and also generalizes well to problems that are up to 10x larger than those experienced in training. They also show that their approach is often faster than competing algorithms, and has very favorable performance/time trade-offs. Overall this work is very impressive and should be published. There's not much more to say.

Reviewer 3



This paper proposes a reinforcement learning framework to learn greedy algorithms which can solve several graph problems, like minimum vertex cover, maximum cut and traveling salesman problem. The underlying contributions of this paper boil down to two points: (1) it provides representations of both graphs and algorithms (2) it provides a way of learning algorithms via reinforcement learning. Strength: 1, The problem this paper aims at, i.e., “how to represent and learn graph algorithms”, is very interesting and important. I think the framework proposed by this paper is still novel given the fact that there are several existing RL based approaches solving similar problems. 2, Authors did extensive experiments to verify that the proposed method can learn a greedy type of algorithm which generalizes well from small sized graphs to large ones under different tasks. 3, The paper is clearly written. Weakness: 1, I feel the title is a bit over-claimed as the paper only focuses on one type of combinatorial algorithm, i.e., greedy algorithm and does not discuss how to generalize beyond it. It would be great to have a title which is more specific. 2, In the comparison experiments, cutting off the time of solvers to 1 hour seems to be problematic as it may introduce noise to the approximation ratios. Given the current graph size (~1000), I would recommend authors use better solvers and let them run to the end. Especially, for MAXCUT problem, some advanced SDP solvers can handle this sized graph in a reasonable amount of time. 3, It would be great to show how the propagation step in the struct2vec model affects the performance on graphs which is not fully connected. Summary: Based on above comments, I would like to recommend for acceptance.